# ON THE GENERALIZATION BOUNDS OF SPIKING NEURAL NETWORKS VIA RADEMACHER COMPLEXITY

## ABSTRACT

Spiking Neural Network (SNN) has garnered increasing attention as one of bio-inspired models due to its great potential in neuromorphic computing and sparse computation. Many algorithms and techniques have been developed; however, theoretical understandings of the generalization, that is, the extent to which SNNs perform well on unseen data, are far from clear. Recently, Zhang et al. (2024) disclosed that the generalization of SNNs with stochastic firing mechanisms can be upper bounded by an exponential function relative to the excitation probability. In this paper, we theoretically investigate the generalization of SNNs with common-used integration-and-fire schemes. We propose the generalization bounds for several LIF expressions via the empirical Rademacher complexity and covering number. Our theoretical results may shed some insight into future studies of SNNs.

## 1 INTRODUCTION

In recent years, Spiking Neural Networks (SNNs) have been attracted increasing attention due to their potential of event-dependent modeling (Yu et al., 2014), neuromorphic computing (Gerstner & Kistler, 2002), and sparse computation (Rozell et al., 2008). The SNN building consists two parts (Zhang et al., 2024), that is, delivering the spike-related information among spiking neurons via connection weights, while converting information between membrane potentials and spike sequences within certain spiking neuron. The inside computations of a spiking neuron usually follows the integration-and-fire paradigm. There has been significant progress on computational and implementation techniques for SNNs in computer vision (Quiroga et al., 2005), speech recognition (Verstraeten et al., 2005; Schrauwen et al., 2008), reinforcement learning (Florian, 2007; Vasilaki et al., 2009), few-short learning (Panda & Roy, 2016; Kheradpisheh et al., 2018), etc. However, theoretical characterizations, especially the generalization, of SNNs are still far from clear.

The generalization depicts whether and to what extent the studied SNN that has been trained on observed spikes performs well on unobserved spike sequences; perhaps, it is the most fundamental concern in artificial intelligence and neuromorphic computing. Despite the emerged efforts on theoretical characterizations, the generalization of SNNs remains mysterious. Recently, Zhang et al. (2024) disclosed that the generalization of SNNs with stochastic firing mechanisms, rather than the conventional integration-and-fire scheme, can be upper bounded by an exponential function relative to the excitation probability. It is potential to reduce the generalization bound of SNNs exponentially by exploiting random algorithms led by stochastic excitation.

In this paper, we theoretically investigate the generalization of SNNs with general spiking neurons. In contrast to the theoretical results of Zhang et al. (2024) that are derived from stochastic firing mechanisms, we here focus on the common-used integration-and-fire schemes and propose a stricter generalization bound of SNNs via the empirical Rademacher complexity and covering number. The main result of this work is listed as follows

**Theorem 1 (Generalization Bound for SNNs)** *Let $\mathcal{H}$ denote the function collection of $L$-layer SNNs with the width of $N_w$ on the time interval $[0, T]$, $N_f$ universally relates to the upper bound of $\|f(\cdot)\|$ for $f \in \mathcal{H}$, $S_n$ indicates the collection of $n$ pairs of training samples, and $L_2(S_n)$ denotes the data-dependent $L^2$ metric space. Assume that the norm value of connection weights is finite, that is, $\|\boldsymbol{w}\| \leq M_w$. Let $\hbar : [-N_f, N_f] \times [-N_f, N_f] \to \mathbb{R}$ denote the non-negative loss function, which satisfies that*

*i)* $\hbar(\cdot,\cdot)$ *is upper bounded by* $M_{\hbar}$, *i.e.,* $\hbar(f,f') \leq M_{\hbar}$ *for all* $f,f' \in [-N_f, N_f]$,

*ii) for any fixed* $f \in [-N_f, N_f]$, *the mapping* $y \mapsto \hbar(f,y)$ *is* $L_{\hbar}$-*Lipschitz for some* $L_{\hbar} > 0$.

*Then with probability at least* $1 - \delta$ *where* $\delta \in (0,1)$, *we have the generalization bound as follows*

$$\mathbb{E}_{(x,y)\sim\mathcal{D}}[\hbar(f(\boldsymbol{w},\mathbf{X}),y)] \leq \frac{1}{n}\sum_{i=1}^{n}\hbar(f(\boldsymbol{w},\mathbf{X}_i,y_i)) + 2L_{\hbar}\,\hat{\mathfrak{R}}_n(\mathcal{H}) + 3M_{\hbar}\sqrt{\frac{\log(2/\delta)}{2n}}\,,$$

*where* $N_f \in \mathcal{O}[(TM_w)^L\,\mathrm{e}^{-TL}]$ *and the empirical Rademacher complexity* $\hat{\mathfrak{R}}_n(\mathcal{H})$ *is bounded by*

$$\hat{\mathcal{R}}_n(\mathcal{H}) \leq \frac{128\,TN_f N_w^{3/2}\log 2}{3\pi n} - 32\sqrt{\frac{2\log 2}{3\pi}}\sqrt{\frac{\alpha\,TN_f N_w^{3/2}}{n}}\,.$$

Theorem 1 shows the generalization bound of SNNs with the typical integration-and-fire scheme, from which the sharp reduction of Rademacher complexity is estimated by a considerably strict bound relative to the covering number. From this theorem, we also observed that the maximum duration $T$ of received spikes sequence, the network width $N_w$, and the network depth $L$ intrinsically determines the generalization performance of SNNs despite the inside computations within spiking neurons.

Table 1: Progresses on generalization bounds and Rademacher complexities of SNNs.

| Studies | Configurations | Approaches | Generalization Bounds |
|---|---|---|---|
| Zhang et al. (2024) | Stochastic Firing | Rademacher Complexity Dropout | $\hat{\mathfrak{R}}_n(\mathcal{H}) \in \mathcal{O}\left(C_w^L\,p_{\max}^{(L+1)/2}\right)$ |
| This work | Common-used Expressions in Eq. (2) | Rademacher Complexity Covering Numbers | $\hat{\mathfrak{R}}_n(\mathcal{H}) \in \mathcal{O}\left(T^{L+1}M_w^L N_w^{3/2}\,\mathrm{e}^{-TL}\right)$ |

Table 1 shows our proposed generalization bound compared to that of Zhang et al. (2024), from which the Rademacher complexity we studied maintains an exponential reduction with respect to the maximum duration $T$ of received spikes sequence and and the network depth $L$, and a polynomial rate relative the network width $N_w$, significantly tighter than the recent advance. Notice that our results are partially consistent with those of Zhang et al. (2024); the excitation probability of a spiking neuron in can be approximately computed by the product of the maximum duration $T$ of the received spike sequence and the empirical firing rate of the whole SNN. This involves an equivalent conversion between the spiking neuron's internal firing mechanism and the external firing rate.

Figure 1 displays the proof sketches of Theorem 1. There are generally three key steps, that is, proving SNN with spiking expressions is of bounded variations in Lemma 3, deriving a stricter bound for covering number in Lemma 4, and bounding the empirical Radermacher complexity from covering number in Lemma 5. Notice that Theorem 1 is proved according to the DEF expression that corresponds to the typical integration-and-fire scheme, but one can also obtain the similar results from other expressions. This proving workflow in Figure 1 also catalyzes the rest organization of this paper. Section 2 introduces the necessary notations. Section 4 completes the proof of Theorem 1 with useful lemmas and terminologies. Section 5 extends the results of Theorem 1 to alternative expressions of spiking neurons. Section 6 conducts experiments to demonstrate the effectiveness of our theoretical results. Section 7 reviews the related studies on theoretical progresses of SNNs. Section 8 concludes this work.

## 2 NOTATIONS

Let $[N] = \{1, 2, \ldots, N\}$ be an integer set for $N \in \mathbb{N}^+$, and $|\cdot|_{\#}$ denotes the number of elements in a collection, e.g., $|[N]|_{\#} = N$.

**Algorithmic Complexity.** Given two functions $g, h\colon \mathbb{N}^+ \to \mathbb{R}$, we denote by $h = \Theta(g)$ if there exist positive constants $c_1, c_2$, and $n_0$ such that $c_1 g(n) \leq h(n) \leq c_2 g(n)$ for every $n \geq n_0$; $h = \mathcal{O}(g)$ if there exist positive constants $c$ and $n_0$ such that $h(n) \leq cg(n)$ for every $n \geq n_0$; $h = \Omega(g)$ if there exist positive constants $c$ and $n_0$ such that $h(n) \geq cg(n)$ for every $n \geq n_0$.

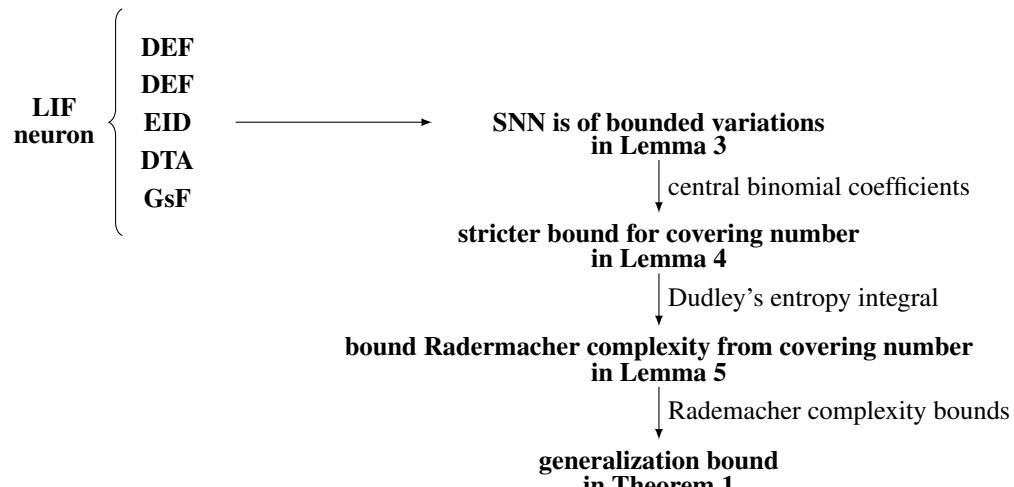

Figure 1: An overview of the proof sketch of Theorem 1.

**Vector and Matrix Norms.** We also consider the two typical norms of vectors or matrices. For $\mathbf{W} \in \mathbb{R}^{n \times m}$, we denote by

$$\|\mathbf{W}\|_2 = \left( \sum_{i=1}^{n} \sum_{j=1}^{m} |\mathbf{W}_{ij}|^2 \right)^{1/2} \quad \text{and} \quad \|\mathbf{W}\|_\infty = \max_{i,j} |\mathbf{W}_{ij}| .$$

Here, we only introduce the norms of $\| \cdot \|_2$ and $\| \cdot \|_\infty$. It is evident that the 2-norm can be bounded by the infinity one, i.e., $\|\boldsymbol{w}\|_2 \le \sqrt{n}\|\boldsymbol{w}\|_\infty$.

**Functional Space.** This work describes the expressive power of neural networks by the Sobolev space and functional norm. Let $f_i$ be a scalar function from $K \subseteq \mathbb{R}^n$ to $\mathbb{R}$. Given $\boldsymbol{\alpha} = (\alpha_1, \alpha_2, \ldots, \alpha_l)^\top \in \mathbb{N}^m$ and $\boldsymbol{x} = (x_1, x_2, \ldots, x_n) \in K$, we define

$$D^{\boldsymbol{\alpha}} f_i(\boldsymbol{x}) = \frac{\partial^{\alpha_1}}{\partial x^{\alpha_1}} \frac{\partial^{\alpha_2}}{\partial x^{\alpha_2}} \cdots \frac{\partial^{\alpha_l}}{\partial x^{\alpha_l}} f_i(\boldsymbol{x}) .$$

We define the space of continuous functions $\mathcal{C}^q(K, \mathbb{R})$ for $q \in \mathbb{N}^+$ by a collection of $f_i$, where $f_i \in \mathcal{C}(K, \mathbb{R})$ and $D^r f_i \in \mathcal{C}(K, \mathbb{R})$ for $r \in [q]$. Let $\mu$ be a Lebesgue measure defined on $K$. Further, we define the Lebesgue spaces for the mapping $f : K \to \mathbb{R}^m$, in which $\mathcal{L}_\mu^p(K, \mathbb{R}^m)$ for $1 \le p < \infty$ and $\mathcal{L}_\mu^\infty(K, \mathbb{R}^m)$ for $p = \infty$, where $f \in \mathcal{C}(K, \mathbb{R}^m)$ and

$$\|f\|_{\mathcal{L}_\mu^p(K, \mathbb{R}^m)} \overset{\text{def}}{=} \left( \int_K \|f(\boldsymbol{x})\|_2^p \, \mathrm{d}\mu(\boldsymbol{x}) \right)^{1/p} < \infty \quad \text{or} \quad \|f\|_{\mathcal{L}_\mu^\infty(K, \mathbb{R}^m)} \overset{\text{def}}{=} \operatorname*{ess\,sup}_{\boldsymbol{x} \in K} \|f(\boldsymbol{x})\|_\infty < \infty .$$

It is evident that $\|f\|_{L_\mu^p(K, \mathbb{R}^m)} \le \sqrt{m\,\mu(K)}\,\|f\|_{L_\mu^\infty(K, \mathbb{R}^m)}$. Let $\mathcal{W}_\mu^{q,p}(K, \mathbb{R}^m)$ denote the Sobolev space as the collection of all functions $f \in \mathcal{C}^q(K, \mathbb{R}^m)$ and $D^{\boldsymbol{\alpha}} f \in \mathcal{L}_\mu^p(K, \mathbb{R}^m)$ for all $|\boldsymbol{\alpha}| \in [q]$.

## 3 SPIKING NEURAL EXPRESSIONS

The computational process of SNNs usually follows the integration-and-firing paradigm, which consists of an integration operation and a firing-reset mechanism. The leaky integration-and-firing (LIF) neuron model is common type of spiking integration operations, of which the general form is as follows

$$\text{DEF:} \quad \tau_{\mathrm{m}} \frac{\mathrm{d}u(t)}{\mathrm{d}t} = -(u(t) - u_{\mathrm{rest}}) + \tau_{\mathrm{r}} f_{\mathrm{agg}}(\boldsymbol{x}(t)) , \tag{1}$$

where $u(t)$ and $u_{\mathrm{rest}}$ separately indicate the membrane and rest potentials of the concerned spiking neuron at timestamp $t$, $\boldsymbol{x}(t) = (x_1(t), \ldots, x_m(t))^\top$ denotes the $m$-dimensional input signals, $\tau_{\mathrm{m}}$ and

$\tau_{\mathrm{r}}$ are positive-valued hyper-parameters with respect to membrane time and membrane resistance, respectively. Here, $f_{\mathrm{agg}}$ is an aggregation function, usually with the form of $f_{\mathrm{agg}}(\boldsymbol{x}(t)) = \boldsymbol{w}^{\top}\boldsymbol{x}(t)$, where $\boldsymbol{w}$ is the learnable vector of connection weights. The spiking neuron fires spikes $s(t)$ at time $t$ if and only if $u(t) \geq u_{\mathrm{firing}}$, where $u_{\mathrm{firing}}$ indicates the firing threshold. Here, we employ the spike excitation function to approximate this procedure, that is, $f_e : \mathbb{R} \to \mathbb{R}$, where $s(t) = f_e(u(t)) = u(t)/u_{\mathrm{firing}}$. After firing, the membrane potential is instantaneously reset to a lower value $u_{\mathrm{reset}}$, that is, reset voltage. Formally, one has $u(t) = (1 - s(t))u(t) + s(t)u_{\mathrm{reset}}$.

Eq. (1) provides a classical formulation of the LIF model via the differential equations, that is, Differential Equation Formulation (DEF). In practice, researchers usually employ other several expressions for the LIF equation, including the Spike Response Model (DEF) scheme (Gerstner, 1995), Exponential-Integral Decomposition (EID) (Stein, 1967), Discrete-Time Approximation (DTA) (Rotter & Diesmann, 1999), Green's Function (GsF) (Gerstner & Kistler, 2002), etc. We list these expressions as follows

$$\begin{cases} \text{SRM: } u(t) = \sum_{\mathrm{f:}\ t^{\mathrm{f}} \leq t} \eta\left(t - t^{\mathrm{f}}\right) + \sum_j w_j \sum_{\mathrm{e:}\ t^{\mathrm{e}}_j \leq t} \epsilon\left(t - t^{\mathrm{e}}_j\right) , \\[2mm] \text{EID: } u(t) = u_{\mathrm{rest}} + (u(t_0) - u_{\mathrm{rest}})\exp\left(-\dfrac{t - t_0}{\tau_{\mathrm{m}}}\right) + \tau_{\mathrm{r}} f_{\mathrm{agg}}(\boldsymbol{x})\left[1 - \exp\left(-\dfrac{t - t_0}{\tau_{\mathrm{m}}}\right)\right] , \\[2mm] \text{DTA: } u[t+1] = u[t] + \dfrac{\Delta t}{\tau_{\mathrm{m}}}\left[-(u[t] - u_{\mathrm{rest}}) + \tau_{\mathrm{r}} f_{\mathrm{agg}}(\boldsymbol{x}[t])\right] , \\[2mm] \text{GsF: } u(t) = u_{\mathrm{rest}} + \displaystyle\int_0^t h(t-s)f_{\mathrm{agg}}(\boldsymbol{x}(s))\,\mathrm{d}s \quad \text{with} \quad h(t) = \dfrac{\tau_{\mathrm{r}}}{\tau_{\mathrm{m}}}\exp\left(\dfrac{-t}{\tau_{\mathrm{m}}}\right) , \end{cases} \tag{2}$$

where $\eta(\cdot)$ denotes the reset kernel following a self-spike, $t^{\mathrm{f}}$ indicates the excitation timing of the concerned spiking neuron, $t^{\mathrm{e}}_j$ indicates the excitation timing of spiking neuron $j$, and $\epsilon(\cdot)$ is the post-synaptic potential kernel. In practice, both $\eta(\cdot)$ and $\epsilon(\cdot)$ are be typically formulated by exponential decay, Gaussian, and piece-wise linear functions, which are usually Lipschitz continuous and have finite derivatives within their domain. Therefore, it is mild to assume that both $\eta(\cdot)$ and $\epsilon(\cdot)$ follow Lipschitz continuity. In the main text, we take DEF in Eq. (1) as the breakthrough point to study the generalization of SNN, and extend the theoretical results and proofs of other expressions in Appendix.

## 4 PROOF OF THEOREM 1

Before the proof, it is necessary to introduce the Gronwall's inequalities, which closely relate to the functions expressed by SNNs with various expressions, and the relation between generalization and Rademacher complexity with some useful terminologies.

### 4.1 BOUNDED VARIATION AND GRONWALL'S INEQUALITIES

We begin our proof with the definition of functions of bounded variation, as the function expressed by SNNs with the aforementioned expressions are observed to exhibit this property.

**Definition 1 (Functions of Bounded Variation)** *The function* $u \in \mathcal{L}^1(\Omega, \mathbb{R})$ *is a function of bounded variation on* $\Omega$*, denoted by* $BV(\Omega, \mathbb{R})$*, if the distributional derivative of* $u$ *is representable by a finite Radon measure in* $\Omega$*, i.e.,*

$$\int_\Omega u \cdot \frac{\partial \varphi}{\partial x_i}\,\mathrm{d}x = -\int_\Omega \varphi\,\mathrm{d}D_i u , \quad \forall\,\varphi \in \mathcal{C}^1(\Omega, \mathbb{R}) , \quad i \in [n] ,$$

*for some Radon measure* $Du = (D_1 u, D_2 u, \ldots, D_n u)$*. We denote by* $|Du|$ *the total variation of the vector measure* $Du$*, i.e.,*

$$|Du|_\Omega = \sup\left\{\int_\Omega u(x)\,\mathrm{div}(\varphi)\,\mathrm{d}x \,\Big|\, \varphi \in \mathcal{C}^1(\Omega, \mathbb{R}^n),\ \|\varphi\|_{\mathcal{L}^\infty(\Omega)} \leq 1\right\} .$$

The above definition refers to Dutta & Nguyen (2018).

Gronwall's inequalities are the key lemmas for connecting the covering number and the function of bounded variants that correspond to SNNs with various expressions. Here, we introduce the discrete (Clark, 1987) and continuous (Howard, 2025) versions as follows.

**Lemma 1 (Discrete Gronwall's Inequality)** *Let $\{u_k\}_{k \geq 0}, \{a_k\}_{k \geq 0}, \{b_k\}_{k \geq 0}$ be positive sequences of real numbers that satisfies $u_n \leq a_n + \sum_{l=0}^{n-1} b_l u_l$ for $n \in \mathbb{N}$, then one has*

$$u_n \leq a_n + \sum_{l=0}^{n-1} a_l b_l \prod_{j=l+1}^{n-1} (1 + b_j), \quad \forall \, n \geq 0.$$

**Lemma 2 (Continuous Gronwall's Inequality)** *Let $R_t$ denote the real-valued interval with forms of $[a, \infty)$ or $[a, b]$ where $a < b$, and $\alpha, \beta, u$ are real-valued functions defined on $R_t$. Assume that (1) both $\beta$ and $u$ are continuous, (2) the negative part of function $\alpha$ is integrable on every closed and bounded subinterval of $R_t$, and (3) function $\beta$ is non-negative and $\alpha$ is non-decreasing. If function $u$ satisfies $u(t) \leq \alpha(t) + \int_a^t \beta(s) u(s) \, \mathrm{d}s$ for $t \in R_t$, then one has*

$$u(t) \leq \alpha(t) \exp \left( \int_a^t \beta(s) \, \mathrm{d}s \right), \quad \forall \, t \in R_t.$$

It is intuitive that the discrete Gronwall's inequality works for the expressions of SRM and DTA, while the continuous Gronwall's inequality works for those of DEF, EID, and GsF.

Based on the above definition and lemmas, we have the first key conclusion as follows.

**Lemma 3** *In the case of finite spikes in $[0, T]$, the function expressed by an SNN with the DEF scheme is the function of bounded variation.*

Lemma 3 shows the well-behaved property of SNNs with the DEF expressions. The proof of Lemma 3 can be accessed from Appendix B.1.

## 4.2 STRICTER BOUND OF COVERING NUMBER

In this subsection, we are going to connect the function of bounded variants and covering number.

**Definition 2 (Covering Number)** *Let $(\mathfrak{M}, \rho)$ be a metric space. A subset $\mathfrak{I} \subseteq \mathfrak{M}$ is called a $\gamma$-cover of $\mathfrak{I} \subseteq \mathfrak{M}$, if for every $m \in \mathfrak{I}$, there exists an $m' \in \hat{\mathfrak{I}}$ such that $\rho(m, m') \leq \gamma$. The $\gamma$-covering number of $\mathfrak{I}$ is defined by $N_{cn}(\gamma, \mathfrak{I}, \rho) = \min\{|\hat{\mathfrak{I}}| : \hat{\mathfrak{I}} \text{ is a } \gamma\text{-cover of } \mathfrak{I}\}$.*

The above definition refers to Bartlett et al. (2017). Thus, we have the second key lemma as follows.

**Lemma 4** *Provided two functional indicator sets*

$$\begin{cases} \mathfrak{I}_{N_w} = \{u \in L^1([0, T]^{N_w}) \mid |u(t)| \text{ is a non-non decreasing function w.r.t. time } t\}, \\ \mathfrak{B}_{N_w} = \{u \in L^1([0, T]^{N_w}) \mid |Du|_{(0,T)^{N_w}} \leq M_u\}, \end{cases}$$

*the collection of spiking computing functions $f(\cdot)$ expressed by an $L$-layer SNN with the DEF expressions can be upper bounded by the covering number, that is,*

$$\begin{cases} N_{cn}(\gamma, \mathfrak{I}_{N_w}, L_2(S_n)) \leq \left[ \dfrac{2^{4(TN-f)\sqrt{N_w}/\gamma}}{6\pi} \right]^{N_w}, \\[4mm] N_{cn}(\gamma, \mathfrak{B}_{N_w}, L_2(S_n)) \leq \left[ \dfrac{2^{16(TN-f)\sqrt{N_w}/\gamma}}{(6\pi)^2} \right]^{N_w}, \end{cases}$$

*where $N_f$ universally relates to the upper bound of $\|f(\cdot)\|$, $S_n$ indicates the collection of $n$ pairs of training samples, and $L_2(S_n)$ denotes the data-dependent $L^2$ metric space.*

Lemma 4 shows a considerably strict bound for the covering number of the functions expressed by SNNs equipped with DEF neurons. The key for proving Lemma 4 comes from an observation that the covering number in Lemma 3 is related to the number of positive integer solutions of the LIF

equation which is equal to central binomial coefficients; the latter obeys a recurrence relation. Thus, it suffices to write the recurrent formation of Eq. (2) as

$$\|f(\boldsymbol{x}(t))\| \le A^L \|\boldsymbol{x}(t)\| + \sum_{l=1}^{L} B A^{L-l-1} = A^L \|\boldsymbol{x}(t)\| + \frac{A^{L-1} - A^{-1}}{A-1} B \stackrel{\text{def}}{=} N_f$$

with

$$A = \frac{t\,\tau_{\text{r}}}{\tau_{\text{m}}\,u_{\text{firing}}} \, \mathrm{e}^{t/\tau_{\text{m}}} \|\boldsymbol{w}\| \quad \text{and} \quad B = \left[ \frac{\|u(0)\|}{u_{\text{firing}}} + \frac{t\,\|u_{\text{rest}}\|}{\tau_{\text{m}}\,u_{\text{firing}}} \right] \exp\left( \frac{t}{\tau_{\text{m}}} \right)$$

and strictly tighten this bound by exploiting the ratio of gamma functions (Gautschi, 1959) as follows

$$x^{1-\lambda} \le \frac{\Gamma(x+1)}{\Gamma(x+\lambda)} \le (x+1)^{1-\lambda} \quad \text{for} \quad x > 0 \quad \text{and} \quad 0 < \lambda < 1 \, .$$

The full proof of Lemma 4 can be accessed from Appendix B.2.

### 4.3 GENERALIZATION AND RADEMACHER COMPLEXITY

In this subsection, we are going to bound the Rademacher complexity by covering number. For simplicity, we here focus on typical classification task, such as the delayed-memory XOR (Abbott et al., 2016) and the spiking sorter (Lee et al., 2017), of which the output is bounded by $[-N_f, N_f]$ or $-[N_f] \cup \{0\} \cup [N_f]$ where $N_f \in \mathbb{N}^+$. Let $\mathcal{W}$ be the connection weight space for SNNs, and $\mathcal{D}$ denotes the underlying joint distribution over input and output space $\mathcal{X} \times \mathcal{Y}$. The training data set $S_n = \{(\mathbf{X}_i, y_i) \in \mathcal{X} \times \mathcal{Y}\}_{i \in [n]}$ is drawn from $\mathcal{D}$. Thus, we establish the function space as $\mathcal{F}_{\mathcal{W}} = \{f(\boldsymbol{w}, \mathbf{X}) \mid \boldsymbol{w} \in \mathcal{W}, \mathbf{X} \in \mathcal{X}\}$. The expected and empirical errors are defined as follows

$$\mathbb{E}_{(x,y) \sim \mathcal{D}}[\hbar(f(\boldsymbol{w}, \mathbf{X}), y)] = \mathbb{E}_{(\mathbf{X}, y)} \left[ \hbar\left(f(\boldsymbol{w}, \mathbf{X}), y\right) \right] \quad \text{and} \quad \widehat{E}(f) = \frac{1}{n} \sum_{i=1}^{n} \hbar\left(f(\boldsymbol{w}, \mathbf{X}_i), y_i\right) \, ,$$

where $\hbar$ denotes the loss function, such as the least square loss and 0-1 loss functions. Here, we mildly assume that the loss function is Lipschitz continuous with respect to $f$, that is, $|\hbar(f, y) - \hbar(f', y)| \le L_\hbar |f - f'|$ for any $f, f'$ that usually are determined by $\boldsymbol{w}$ and $\mathbf{X}$.

The generalization describes the gap between $\mathbb{E}_{(x,y) \sim \mathcal{D}}[\hbar(f(\boldsymbol{w}, \mathbf{X}), y)]$ and $\widehat{E}(f)$. Rademacher complexity that measures how well a class of functions can fit random noise is a typically statistical concept for deriving generalization bounds (Mohri et al., 2018).

**Definition 3 (Rademacher Complexity)** *Given a class of functions $\mathcal{H}$ mapping from $\mathcal{X}$ to $\mathbb{R}$ and a sample $S_n = \{\mathbf{X}_1, \ldots, \mathbf{X}_n\}$ drawn from a distribution $\mathcal{D}$, the empirical Rademacher complexity of $\mathcal{H}$ with respect to $S_n$ is*

$$\hat{\mathfrak{R}}_n(\mathcal{H}) = \mathbb{E}_\sigma \left[ \sup_{f \in \mathcal{H}} \frac{1}{n} \sum_{i=1}^{n} \sigma_i f(\mathbf{X}_i) \right] ,$$

*where $\sigma_i$ are independent Rademacher variables that take values $\pm 1$ with equal probability, and the expectation $\mathbb{E}_\sigma$ is taken over their distribution.*

Next, we are going to introduce the typical lemma as follows.

**Lemma 5** *Let $\mathcal{H}$ be the hypothesis space, and $L^2(S_n)$ denotes the data-dependent $L^2$ metric space. For $f \in \mathcal{H}$, we have*

$$\hat{\mathfrak{R}}_n(\mathcal{H}) \le \inf_{\epsilon \ge 0} \left\{ 4\epsilon + \frac{12}{\sqrt{n}} \int_\epsilon^{\sup_{f \in \mathcal{H}} \sqrt{\mathbb{E}[f^2]}} \sqrt{\log N_{cn}(\gamma, \mathcal{H}, L^2(S_n))} \, \mathrm{d}\tau \right\} \, .$$

Lemma 5 provides a typical bridge for bounding the Rademacher complexity by covering number. The details can refer to the note of Srebro & Sridharan (2010).

**Finishing the proof of Theorem 1.** From Lemma 4, we have

$$\sqrt{\log N_{\text{cn}}(\gamma, \mathfrak{B}_{N_w}, L_2(S_n))} \le \sqrt{\frac{8\,T N_f N_w^{3/2} \log 2}{3\pi\gamma}} \stackrel{\text{def}}{=} g(\gamma) \, .$$

Thus, we have

$$\int_a^b g(\gamma)\,\mathrm{d}\gamma = \sqrt{\frac{32\log 2}{3\pi}}\sqrt{TN_f N_w^{3/2}}\left[\sqrt{b}-\sqrt{a}\right]\,. \tag{3}$$

Inserting the above result into Lemma 5, we have

$$\hat{\mathfrak{R}}_n(\mathfrak{B}_{N_w}) \le \inf_{\epsilon\ge 0}\left\{4\epsilon + 12\int_\epsilon^\alpha \sqrt{\frac{\log N_{\mathrm{cn}}(\gamma,\mathfrak{B}_{N_w},L^2(S_n))}{n}}\,\mathrm{d}\tau\right\}\,.$$

where $\alpha = \sup_{f\in\mathfrak{B}_{N_w}}\sqrt{\mathbb{E}[f^2]}$. By exploiting Eq. (3), it is observed that

$$\hat{\mathcal{R}}_n(\mathfrak{B}_{N_w}) \le \inf_{\epsilon\ge 0}\left\{4\epsilon + 32\sqrt{\frac{2\log 2}{3\pi}}\sqrt{\frac{TN_f N_w^{3/2}}{n}}\left(\sqrt{b}-\sqrt{\epsilon}\right)\right\}\,.$$

This implies that

$$\hat{\mathcal{R}}_n(\mathfrak{B}_{N_w}) \le \frac{128\,TN_f N_w^{3/2}\log 2}{3\pi n} - 32\sqrt{\frac{2\log 2}{3\pi}}\sqrt{\frac{\alpha\,TN_f N_w^{3/2}}{n}}\,. \tag{4}$$

According to the Rademacher complexity regression bounds of Mohri et al. (2018), one can bound the generalization relative to above spiking sorter task by Rademacher complexity as follows

**Lemma 6** *Let $\mathcal{H} = \{f : \mathcal{X} \to \mathcal{Y}\}$ be a set of functions. Let $\hbar : \mathcal{Y}\times\mathcal{Y} \to \mathbb{R}^+$ be a non-negative loss function upper bounded by $M_\hbar > 0$, and $L_\hbar$-Lipschitz in the second variable. Then with probability at least $1 - \delta$ where $\delta \in (0,1)$, the following holds*

$$\mathbb{E}_{(x,y)\sim\mathcal{D}}[\hbar(f(\boldsymbol{w},x),y)] \le \frac{1}{n}\sum_{i=1}^n \hbar(f(\boldsymbol{w},\mathbf{X}_i,y_i)) + 2L_\hbar\,\hat{\mathfrak{R}}_n(\mathcal{H}) + 3M_\hbar\sqrt{\frac{\log(2/\delta)}{2n}}\,.$$

By inserting Eq. (4) into Lemma 6 and $\mathcal{H} \subseteq \mathfrak{B}_{N_w}$, we can finish the proof of Theorem 1. $\qquad\square$

## 5 EXTENSION RESULTS TO OTHER LIF EXPRESSIONS

Based on the main conclusion in Theorem 1 and its proof, we can extend the results to other expressions in Eq. (2). According to the Gronwall's inequalities with discrete (Clark, 1987) and continuous (Howard, 2025) versions in Section 4, we divide the LIF expressions in Eq. (2) into two categories; the first one that contains EID and GsF employ the discrete Gronwall's inequality, almost same as that of DEF, while the second one that comprises SRM and DTA obeys the continuous Gronwall's inequality. We straightway list the theoretical results as follows

**Theorem 2 (Generalization Bound for Various LIF Expressions)** *Let $T$ is the maximum duration of received spikes sequence, $N_w$ denotes the network width, $N_f$ universally relates to the upper bound of $\|f(\cdot)\|$ for $f$ belonging to the hypothesis space $\mathcal{H}$, $S_n$ indicates the collection of $n$ pairs of training samples, and $L_2(S_n)$ denotes the data-dependent $L^2$ metric space. Assume that the norm value of connection weights is finite, that is, $\|\boldsymbol{w}\| \le M_w$. Let $\hbar : [-N_f,N_f]\times[-N_f,N_f] \to \mathbb{R}$ denote the non-negative loss function, which satisfies that*

*i) $\hbar(\cdot,\cdot)$ is upper bounded by $M_\hbar$, i.e., $\hbar(f,f') \le M_\hbar$ for all $f,f' \in [-N_f,N_f]$,*

*ii) for any fixed $f \in [-N_f,N_f]$, the mapping $y \mapsto \hbar(f,y)$ is $L_\hbar$-Lipschitz for some $L_\hbar > 0$.*

*Then for each LIF expressions in Eq. (2) and the function collection $\mathcal{H}$ of L-layer SNNs on the time interval $[0,T]$, with probability at least $1 - \delta$ where $\delta \in (0,1)$, the following bound holds*

$$\mathbb{E}_{(x,y)\sim\mathcal{D}}[\hbar(f(\boldsymbol{w},\mathbf{X}),y)] \le \frac{1}{n}\sum_{i=1}^n \hbar(f(\boldsymbol{w},\mathbf{X}_i,y_i)) + 2L_\hbar\,\hat{\mathfrak{R}}_n(\mathcal{H}) + 3M_\hbar\sqrt{\frac{\log(2/\delta)}{2n}}\,,$$

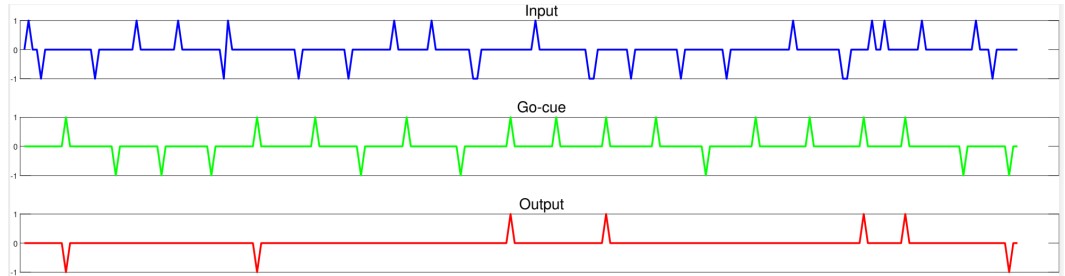

Figure 2: Illustrations of the delayed-memory XOR task, where the panels from top to bottom are the single-trial input, go-cue signals, and output traces, respectively.

*in which the empirical Rademacher complexity $\hat{\mathfrak{R}}_n(\mathcal{H})$ is bounded by*

$$\hat{\mathcal{R}}_n(\mathcal{H}) \leq \frac{128\,T N_f N_w^{3/2} \log 2}{3\pi n} - 32\sqrt{\frac{2\log 2}{3\pi}}\sqrt{\frac{\alpha\,T N_f N_w^{3/2}}{n}}$$

*and the covering number estimated is of forms*

$$N_f^c \in \mathcal{O}[(TM_w)^L\,\mathrm{e}^{-TL}] \quad and \quad N_f^d \in \mathcal{O}[(TM_w)^{L-1}\,\mathrm{e}^{-T(L-1)}],$$

*where $N_f^c$ universally relates to the upper bound of the SNN function with the EID and GsF expressions, while $N_f^d$ indicates those of the SRM and DTA expressions.*

It is observed that using various expression form of LIF spiking neurons does not influence the algorithmic complexity of the SNN generalization bound; only some constants are different. The proof of Theorem 2 can be accessed from Appendix C.

## 6 EXPERIMENTS

In this section, we conducted simulation experiments to evaluate the effectiveness of our theoretical results, especially about the influence of the maximum duration $T$ of received spikes sequence, the network width $N_w$, and the network depth $L$ on the generalization performance of SNNs.

Here, we consider the *delayed-memory XOR* task, which performs the XOR operation on the input history stored over an extended duration (Abbott et al., 2016). Specifically, the network receives two binary pulse signals, + or -, through an input channel and a go-cue channel. When the network receives two input pulses between two go-cue pulses, it should output the XOR signal of both inputs. In other words, the network outputs a positive signal if the input pulses are of equal signs (+ + or - -), and a negative signal if the input pulses are of opposite signs (+ - or - +). If there is only one input pulse between two go-cue pluses, the network should generate a null output. Here, we simulated a Delayer-memory XOR dataset, which consists of 300 input pulses, 200 go-cue pulses, and the corresponding output signals in the time interval $[0, T]$. We also train SNNs with the rest voltage $u_{\mathrm{rest}} = 0$ by the first 80% timestamps and predict the output signals of the last 20% timestamps.

Here, we employ SNNs with the SRM expressions and take values of $T = [500 : 500 : 3000]$, $N_w = [2 : 2 : 8]$, and $\log_2 L = [1 : 1 : 4]$. The generalization bound is empirically measured by the gap between testing and training errors, that is, $\epsilon$ = testing error - training error. We also test 10 trials for counting the expectation generalization and its variance. All models are implemented via SpikingJelly (Fang et al., 2023) and conducted on Intel i9-12900K.

Figure 2 displays the curves of the generalization performance of SNNs relative to the maximum duration $T$ of received spikes sequence, the network width $N_w$ and the network depth $L$. There are three observations: (1) there exists a linearly negative correlation between $\epsilon$ and $\log_2 L$; (2) $N_w$ is negatively correlated with $\epsilon$ at an almost linear rate; (3) the effect of $T$ on $\epsilon$ is not obvious once $T \geq 500$. It is concluded that observation (1) obviously coincides with the results of Theorem 1, while our theoretical analysis is still relatively loose in explaining observations (2) and (3). These investigated results confirm the effectiveness of our theoretical results.

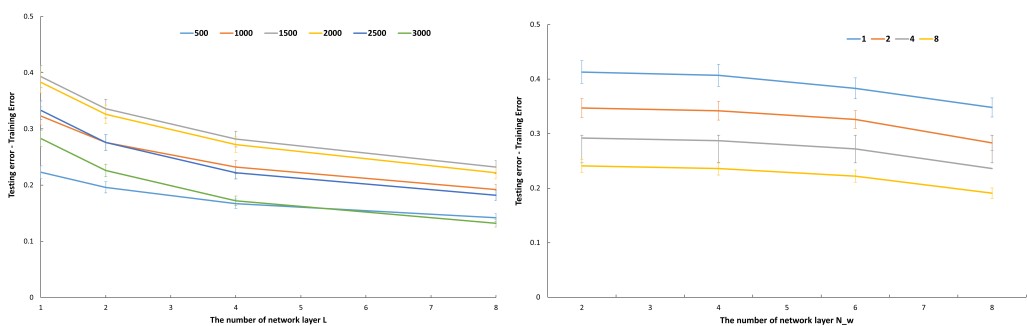

Figure 3: The curves of the generalization performance $\epsilon$ of SNNs relative to (left) the case of the maximum duration $T$ of received spikes sequence and the network depth $L$ provided $N_w = 4$ and (right) the case of the network width $N_w$ and the network depth $L$ provided $T = 2000$.

## 7 RELATED STUDIES

There have been several efforts on examining the universality of SNNs, which showd that the designed SNNs or spiking neural P systems can simulate some typical computational models, involving Turing machines (Maass & Bishop, 2001), random access machines (Maass, 1997), threshold circuits (Maass, 1996; Maass & Markram, 2004), propagation paths (She et al., 2021), dynamical systems (Zhang et al., 2021; Zhang & Zhou, 2022), etc. There are few academic studies on the computational efficiency of SNNs for some specific issues, such as the convergence in the limit results for the sparse coding problem (Tang, 2016; Tang et al., 2017), the computational complexity of SNNs for temporal quadratic programming (Barrett et al., 2013; Chou et al., 2019), and the time complexity of approximating multivariate spike flows (Zhang & Zhou, 2022). Besides, the firing rates or equally the number of firing spikes are the alternative measure of network activities for investigating neural computation and model dynamics because of the close relation between firing rates and network function, including neural input, connectivity, spiking function, and firing process (Adrian, 1926; Aertsen et al., 1980). The averaged firing rate is used to approximate the optimal solutions of some quadratic programs within polynomial complexity (Chou et al., 2019), while the instantaneous firing rate is used to ensure that SNNs can approximate dynamical systems well (Zhang & Zhou, 2022).

Despite the emerged efforts on theoretical characterizations, it remains mysterious whether and to what extent the studied SNN that has been trained on observed spikes performs well on unobserved spike sequences, which is the most fundamental concern in artificial intelligence and neuromorphic computing. A recent advance (Zhang et al., 2024) disclosed that the generalization of SNNs with stochastic firing mechanisms, rather than the conventional integration-and-fire scheme, can be upper bounded by an exponential function relative to the excitation probability, which implies a way to reduce the generalization bound of SNNs exponentially by exploiting random algorithms led by stochastic excitation.

## 8 CONCLUSIONS

In this paper, we theoretically investigated the generalization of SNNs with the common-used integration-and-fire schemes and proposed a generalization bound of SNNs via the empirical Rademacher complexity and covering number. In contrast to the theoretical results of Zhang et al. (2024) that are derived from stochastic firing mechanisms, our results take a considerably stricter generalization bound and implied that the maximum duration $T$ of received spikes sequence, the network width $N_w$, and the network depth $L$ intrinsically determines the generalization performance of SNNs despite the inside computations within spiking neurons. Numerical experiments demonstrate the effectiveness of our theoretical results.

## ETHICS STATEMENT

The research presented in this paper is purely computational and focuses on the theoretical development of an algorithmic model. All experiments were conducted on publicly available and anonymized datasets, which do not contain any personally identifiable or sensitive information. Our work does not involve human subjects, animal testing, or confidential data. To the best of our knowledge, we foresee no direct negative ethical implications or societal consequences resulting from this research.

## REPRODUCIBILITY STATEMENT

To ensure the reproducibility of our work, this paper fully discloses all necessary details to replicate the main experimental results. The experiments in this paper are intended to verify the effectiveness of the theory, so mature algorithms and configurations are used. Section Experiments provides a comprehensive description of the training and testing procedures, including dataset specifications and splits, all hyperparameter values and their selection process, optimizer types, and other relevant configuration settings. While the source code is not publicly available at the time of submission, we believe the details provided are sufficient for independent reproduction of our findings. We commit to releasing the complete source code publicly upon acceptance of the paper.

## USE OF LARGE LANGUAGE MODELS

This paper was written without the assistance of any large language model or intelligent agents.

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
