## A    APPENDIX

This appendix provides the supplementary materials for this work, constructed according to the corresponding sections therein. For convenience, we here take both $\tau_m$ and $\tau_r$ as positive values, in order to avoid the redundance led by $|\tau_m|$ and $|\tau_r|$.

## B    PROOFS OF LEMMAS RELATIVE TO THEOREM 1

This section provides the proofs for three Lemmas [3, 4, 5], which we used to prove Theorem 1.

### B.1    PROOF OF LEMMA 3

Recall the DEF scheme, that is,

$$\tau_m \frac{du(t)}{dt} = -(u(t) - u_{\text{rest}}) + \tau_r f_{\text{agg}}(\boldsymbol{x}(t)) \ .$$

For any $t_1, t_2 \in [T]$, we have

$$|u(t_1) - u(t_2)| \leq \tau_m \left| \frac{du(t_1)}{dt} - \frac{du(t_2)}{dt} \right| + \tau_r |f_{\text{agg}}(\boldsymbol{x}(t_1)) - f_{\text{agg}}(\boldsymbol{x}(t_2))| \ .$$

According to the Picard-Lindelof theorem, the membrane potential $u(t)$ in the DEF expression exists uniquely and is absolutely continuous. Let $M_{\text{agg}}$ denote the maximum norm of the aggregation function $f_{\text{agg}}(\cdot)$, that is, $|f_{\text{agg}}(\boldsymbol{x}(t))| \leq M_{\text{agg}}$. Thus, one has

$$\left| \frac{du(t_1)}{dt} - \frac{du(t_2)}{dt} \right| \leq M_{\text{agg}} |t_1 - t_2| \ .$$

According to Subsection 3, the aggregation function $f_{\text{agg}}(\cdot)$ is linear and thus Lipschitz continuous, i.e., there exist a constant $L_{\text{agg}}$ such that $|f_{\text{agg}}(\boldsymbol{x}(t_1)) - f_{\text{agg}}(\boldsymbol{x}(t_2))| \leq L_{\text{agg}} |t_1 - t_2|$. Therefore, we conclude that the membrane potential $u(t)$ in the DEF expression is Lipschitz continuous with constant $L_u$

$$|u(t_1) - u(t_2)| \leq \tau_m M_{\text{agg}} |t_1 - t_2| + \tau_r L_{\text{agg}} |t_1 - t_2| \leq L_u |t_1 - t_2| \ ,$$

where $L_u = \tau_m M_{\text{agg}} + \tau_r L_{\text{agg}}$.

For any partition $0 = t_0 < t_1 < \cdots < t_n = T$, one has

$$u(t_i) - u(t_{i-1}) = \frac{d}{dt}(v_i) \cdot (t_i - t_{i-1}) \quad \text{for some } v_i \in (t_{i-1}, t_i) \ ,$$

according to the Mean Value Theorem. By summing up the absolute differences that gives the total variation, we have

$$\sum_{i=1}^{n} |u(t_i) - u(t_{i-1})| = \sum_{i=1}^{n} |v_i| (t_i - t_{i-1}) \ .$$

It is observed that $u(t)$ is bounded, i.e., $|u(t)| \leq M_u$. Thus, the total variation can be bounded by

$$V_0^T(u) \leq M_u \sum_{i=1}^{n} (t_i - t_{i-1}) = M_u T \ .$$

Consider the spike excitation function, we also can conclude that

$$|s(t_1) - s(t_2)| = |f_e(u(t_1)) - f_e(u(t_2))| \leq \frac{u(t_1) - u(t_2)}{u_{\text{firing}}} \ .$$

It is evident that both $u(t)$ and $s(t)$ have finite total variation due to $M_u T < \infty$ and $M_u T / u_{\text{firing}} < \infty$. Therefore, the function expressed by single spiking neuron with the DEF expression is of bounded variation. Since connection weights is independent to $t$ and bounded by $M_w$, we can further conclude that the function expressed by an SNN with the DEF expression is the function of bounded variation. $\square$ This completes the proof.

## B.2 PROOF OF LEMMA 4

We start this proof with the case of $N_w = 1$. Let $Du$ denote the distributional derivative of the membrane potential $u(t)$ in the DEF expression and

$$
\begin{cases}
\mathfrak{I} = \{u \in L^1([0,T]) \mid u(t) \text{ is a non-non decreasing function w.r.t. time } t\} \,, \\
\mathfrak{B} = \{u \in L^1([0,T]) \mid |Du|_{(0,T)} \le M_u\} \,.
\end{cases}
$$

From the conversion of Zhang et al. (2021), the membrane potential $u(t)$ in the DEF expression is evidently equivalent to finding solution to the following equation if $u(t)$ is Lipschitz continuous

$$
u(t) = u(0) + \frac{1}{\tau_{\mathrm{m}}} \int_0^t -(u(\tau) - u_{\mathrm{rest}}) + \tau_{\mathrm{r}} f_{\mathrm{agg}}(\boldsymbol{x}(\tau)) \, \mathrm{d}\tau \,.
$$

By taking norms, this yields

$$
\|u(t)\| = \|u(0)\| + \frac{1}{\tau_{\mathrm{m}}} \int_0^t \|u(\tau)\| + \|u_{\mathrm{rest}} + \tau_{\mathrm{r}} f_{\mathrm{agg}}(\boldsymbol{x}(\tau))\| \, \mathrm{d}\tau
$$

$$
\le \|u(0)\| + \frac{1}{\tau_{\mathrm{m}}} \int_0^t \|u(\tau)\| \, \mathrm{d}\tau + \frac{t}{\tau_{\mathrm{m}}} \|u_{\mathrm{rest}} + \tau_{\mathrm{r}} f_{\mathrm{agg}}(\boldsymbol{x}(t))\| \quad (\text{ inserting } \int_0^t \mathrm{d}\tau = t )
$$

According to the continuous Gronwall's inequality in Lemma 2, we have

$$
\|u(t)\| \le \left[\|u(0)\| + \frac{t}{\tau_{\mathrm{m}}} \|u_{\mathrm{rest}} + \tau_{\mathrm{r}} f_{\mathrm{agg}}(\boldsymbol{x}(t))\|\right] \exp\left(\frac{1}{\tau_{\mathrm{m}}} \int_0^t \mathrm{d}\tau\right)
$$

$$
\le \left[\|u(0)\| + \frac{t}{\tau_{\mathrm{m}}} \|u_{\mathrm{rest}} + \tau_{\mathrm{r}} f_{\mathrm{agg}}(\boldsymbol{x}(t))\|\right] \exp\left(\frac{t}{\tau_{\mathrm{m}}}\right) \tag{5}
$$

Provided the $L$-layer SNN of the following form[1]

$$
\begin{cases}
f(\boldsymbol{x}(t)) = f_e(u^{(L)}(t)) \,, \\
\boldsymbol{s}^{(l)}(t) = f_e(u^{(l)}(t)) \,, \\
u^{(l)}(t) \leftarrow \mathrm{DEF}\left[u^{(l)}(t-1), \boldsymbol{w}^\top \boldsymbol{s}^{(l-1)}(t)\right] \,, \\
\boldsymbol{s}^{(0)}(t) = \boldsymbol{x}(t) \,,
\end{cases}
$$

the norm of the expressive function can be unfolded as

$$
\|f(\boldsymbol{x}(t))\| = \|f_e(u^{(L)}(t)) - f_e(0)\| \le \frac{1}{u_{\mathrm{firing}}} \|u^{(L)}(t) - 0\|
$$

$$
\le \frac{1}{u_{\mathrm{firing}}} \left[\|u(0)\| + \frac{t}{\tau_{\mathrm{m}}} \|u_{\mathrm{rest}} + \tau_{\mathrm{r}} \boldsymbol{w}^\top \boldsymbol{s}^{(l-1)}(t)\|\right] \exp\left(\frac{t}{\tau_{\mathrm{m}}}\right) \quad (\text{ inserting Eq. (5) })
$$

$$
\le \left[\frac{\|u(0)\|}{u_{\mathrm{firing}}} + \frac{t\|u_{\mathrm{rest}}\|}{\tau_{\mathrm{m}} u_{\mathrm{firing}}}\right] \exp\left(\frac{t}{\tau_{\mathrm{m}}}\right) + \frac{t\,\tau_{\mathrm{r}}}{\tau_{\mathrm{m}} u_{\mathrm{firing}}} \mathrm{e}^{t/\tau_{\mathrm{m}}} \|\boldsymbol{w}\| \|\boldsymbol{s}^{(l-1)}(t)\| \,.
$$

Next, we introduce a useful lemma relative the Gronwall's inequality (Verma & Kumar, 2025).

**Lemma 7** *Let $(u_k)_{k\ge0}$ be a sequence that satisfies $u_k \le a_k u_{k-1} + b_k$ for all $k \ge 1$, where $(a_k)_{k\ge1}, (b_k)_{k\ge1}$ are two positive sequences. Then it holds*

$$
u_k \le \left(\prod_{j=1}^k a_j\right) u_0 + \sum_{j=1}^k b_j \left(\prod_{i=j+1}^k a_i\right) \quad \text{for all} \quad k \ge 1 \,.
$$

According to Lemma 7, we can further bound the norm of the expressive function by

$$
\|f(\boldsymbol{x}(t))\| \le A^L \|\boldsymbol{x}(t)\| + \sum_{l=1}^L B A^{L-l-1} = A^L \|\boldsymbol{x}(t)\| + \frac{A^{L-1} - A^{-1}}{A-1} B \,, \tag{6}
$$

---

[1]Here, the superscript indicates the layer. But we omit the superscript of connection weights $\boldsymbol{w}$ for simplicity.

where

$$A = \frac{t\,\tau_{\mathrm{r}}}{\tau_{\mathrm{m}}\,u_{\mathrm{firing}}} \exp\left(\frac{t}{\tau_{\mathrm{m}}}\right) \|\boldsymbol{w}\| \quad \text{and} \quad B = \left[\frac{\|u(0)\|}{u_{\mathrm{firing}}} + \frac{t\,\|u_{\mathrm{rest}}\|}{\tau_{\mathrm{m}}\,u_{\mathrm{firing}}}\right] \exp\left(\frac{t}{\tau_{\mathrm{m}}}\right) .$$

Here, we employ $N_f$ to upper bound $\|f(\boldsymbol{x}(t))\|$. Provided that $\|\boldsymbol{w}\| \leq M_w$ and $\|\boldsymbol{x}(t)\| \leq M_x \approx 1$, we can intuitively force that

$$N_f = \tilde{A}^L M_x + \frac{\tilde{A}^L - 1}{\tilde{A}(\tilde{A} - 1)} \tilde{B}$$

with

$$\tilde{A} = \frac{T\,\tau_{\mathrm{r}}}{\tau_{\mathrm{m}}\,u_{\mathrm{firing}}} \exp\left(\frac{T}{\tau_{\mathrm{m}}}\right) M_w \quad \text{and} \quad \tilde{B} = \left[\frac{\|u(0)\|}{u_{\mathrm{firing}}} + \frac{T\,\|u_{\mathrm{rest}}\|}{\tau_{\mathrm{m}}\,u_{\mathrm{firing}}}\right] \exp\left(\frac{T}{\tau_{\mathrm{m}}}\right) .$$

Therefore, we can conclude that $N_f \in \mathcal{O}[(TM_w)^L \exp(-TL)]$, from which

$$\max_{T} \quad N_f(T, L) \in \mathcal{O}\left(\mathrm{e}^{-L}\right) ,$$

$$N_f(T, L) \to 0 \quad \text{as} \quad T \to 0^+ \quad \text{or} \quad T \to +\infty ,$$

and

$$N_f(T, L) \to 0 \quad \text{as} \quad T \to +\infty \quad \text{with an exponential ratio} .$$

Next, we proceed to compute $N_{\mathrm{cn}}(\gamma, \mathfrak{I}, L_2(S_n))$. The proof line follows that of (Verma & Kumar, 2025). For a fixed positive integer $N$, let us set the discretization size as $\Delta x = T/N$, $\Delta y = N_f/N$. To each $z \in \mathfrak{I}$, we associate the pair of functions $(\psi^+[z], \psi^-[z])$ defined by

$$\psi^+[z] = \sum_{k=0}^{N-1} \psi_k^+ \cdot \mathbf{1}[k \cdot \Delta x, (k+1) \cdot \Delta x] ,$$

where

$$\psi_k^- = \left\lfloor \frac{z(k \cdot \Delta x + 0)}{\Delta y} \right\rfloor \quad \text{and} \quad \psi_k^+ = \left\lfloor \frac{z((k+1) \cdot \Delta x - 0)}{\Delta y} \right\rfloor + 1 .$$

For $\mathcal{X}^- \leq \mathcal{X}^+ \in \mathfrak{I}$, one defines $U(\mathcal{X}^-, \mathcal{X}^+) = \{z \in \mathfrak{I} \mid \mathcal{X}^- \leq z \leq \mathcal{X}^+\}$. It is easily proved that the set $\mathcal{U} = \{U(\mathcal{X}^-[z], \mathcal{X}^+[z]) \mid f \in \mathcal{I}\}$ is a covering of $\mathfrak{I}$ due to $z \in U(\mathcal{X}^-[z], \mathcal{X}^+[z])$.

According to

$$\begin{aligned}
\#\mathcal{U} &\leq \{0 \leq a_0 \leq a_1 \leq \cdots \leq a_{N-1} \leq N \mid (a_k \in \mathbb{N})\}^2 \\
&\leq \{(p_1, \ldots, p_{N+1}) \in \mathbb{N}^{N+1} \mid p_1 + \cdots + p_{N+1} = N\}^2 \\
&\leq \binom{2N}{N}^2 ,
\end{aligned}$$

the covering number for the class of functions in $\mathfrak{I}$ is bounded by $\binom{2N}{N}^2$. Consider sums of powers of binomial coefficients

$$a_n^{(r)} = \sum_{k=0}^{n} \binom{n}{k}^r .$$

For $r = 2$, the closed-form solution is given by

$$a_n^{(2)} = \binom{2n}{n} .$$

This implies that the central binomial coefficients $a_n^{(2)}$ obeys the recurrence relation

$$(n+1)a_{n+1}^{(2)} - (4n+2)a_n^{(2)} = 0 .$$

By solving the aforementioned upper bound of $\#\mathcal{U}$, we have

$$\binom{2N}{N} = C_1 \frac{4^{N-1}}{\Gamma(N+1)} \left(\frac{3}{2}\right)_{2N-1} \quad (\text{ here, } ((x))_N \text{ denotes the Pochhammer symbol })$$

$$= 2 \cdot \frac{2^{2(N-1)}}{\Gamma(N+1)} \left(\frac{3}{2}\right)_{2N-1} \quad (\text{ let } C_1 = 2 )$$

$$= \frac{2^{2(N-1)}}{\Gamma(N+1)} \frac{\Gamma(\frac{3}{2}+n-1)}{\Gamma(\frac{3}{2})} = \frac{2^{2(N-1)}}{\Gamma(N+1)} \Gamma\left(N+\frac{1}{2}\right) \frac{\sqrt{\pi}}{2}$$

$$= \frac{2^{2N}}{\sqrt{\pi}} \frac{\Gamma(N+\frac{1}{2})}{\Gamma(N+1)}$$

$$\leq \frac{2^{2N}}{\sqrt{\pi}} \frac{1}{\sqrt{N}} \quad (\text{ from the ratio of gamma functions (Gautschi, 1959) })$$

$$= \frac{2^{2N}}{\sqrt{\pi N}} .$$

Hence, we can conclude that

$$\binom{2N}{N}^2 \leq \frac{2^{4N}}{\pi N} \leq \frac{2^{4N}}{6\pi} ,$$

where the second inequality holds from $N \geq 6$. Let $N = \lceil (TN-f)/\gamma \rceil + 1$, then

$$N_{\mathrm{cn}}(\gamma, \mathfrak{I}, L_2(S_n)) \leq \frac{2^{4(TN-f)/\gamma}}{6\pi} .$$

From the bound proposed by Dutta & Nguyen (2018), that is,

$$N_{\mathrm{cn}}\left(\gamma, \mathfrak{B}, L^2(S_n)\right) \leq N_{\mathrm{cn}}^2\left(\gamma/2, \mathfrak{I}, L^2(S_n)\right) ,$$

a stricter bound is proved by

$$N_{\mathrm{cn}}(\gamma, \mathfrak{B}, L_2(S_n)) \leq \frac{2^{16(TN-f)/\gamma}}{(6\pi)^2} .$$

The above computations can be easily extended to the case of $N_w \geq 1$ where all variables are still bounded by vector or matrix norms. For $u(t) \in \mathbb{R}^{N_w}$, we have

$$\begin{cases} N_{\mathrm{cn}}(\gamma, \mathfrak{I}_{N_w}, L_2(S_n)) \leq \left[\dfrac{2^{4(TN-f)\sqrt{N_w}/\gamma}}{6\pi}\right]^{N_w} , \\[3ex] N_{\mathrm{cn}}(\gamma, \mathfrak{B}_{N_w}, L_2(S_n)) \leq \left[\dfrac{2^{16(TN-f)\sqrt{N_w}/\gamma}}{(6\pi)^2}\right]^{N_w} . \end{cases}$$

This completes the proof. $\qquad\square$

## C  PROOFS AND USEFUL LEMMAS OF THEOREM 2

This section provides the proofs for Theorem 2. The results of other LIF expressions follows closely the proof of Theorem 1. Hence, we here only show the computational difference led by the SRM and DTA expressions. We begin the proof by taking the SRM expression as an example.

**Lemma 8** *In the case of finite spikes in* $[0, T]$*, the function expressed by an SNN with the SRM scheme is the function of bounded variation.*

*Proof.*    Recall the SRM scheme, that is,

$$u(t) = \sum_{\mathrm{f}:\, t^{\mathrm{f}} \leq t} \eta\left(t - t^{\mathrm{f}}\right) + \sum_j w_j \sum_{\mathrm{e}:\, t_j^{\mathrm{e}} \leq t} \epsilon\left(t - t_j^{\mathrm{e}}\right) .$$

According to Subsection 3, the kernels $\eta(\cdot)$ and $\epsilon(\cdot)$ are Lipschitz continuous, i.e., there exist constants $L_\eta$ and $L_\epsilon$ such that

$$|\eta(t_1) - \eta(t_2)| \le L_\eta |t_1 - t_2| \quad \text{and} \quad |\epsilon(t_1) - \epsilon(t_2)| \le L_\epsilon |t_1 - t_2| \,.$$

For any $t_1, t_2 \in [T]$, we have

$$|u(t_1) - u(t_2)| \le \sum_{\text{f: } t^f < t} \left| \eta\left(t_1 - t^f\right) - \eta\left(t_2 - t^f\right) \right| + \sum_j \sum_{\text{e: } t^e_j < t} \left| \epsilon\left(t_1 - t^e_j\right) - \epsilon\left(t_2 - t^e_j\right) \right| \,.$$

Consider a finite number of spikes $N_f$ and $N_e$, the above inequality can be written by

$$|u(t_1) - u(t_2)| \le N_f L_\eta |t_1 - t_2| + N_e L_\epsilon |t_1 - t_2| = L_u |t_1 - t_2| \,,$$

where $L = N_f L_\eta + N_e L_\epsilon$. Thus, we can conclude that the SRM function is Lipschitz continuous.

For any partition $0 = t_0 < t_1 < \cdots < t_n = T$, one has

$$u(t_i) - u(t_{i-1}) = \frac{\mathrm{d}}{\mathrm{d}t}(v_i) \cdot (t_i - t_{i-1}) \quad \text{for some } v_i \in (t_{i-1}, t_i) \,,$$

according to the Mean Value Theorem. By summing up the absolute differences that gives the total variation, we have

$$\sum_{i=1}^n |u(t_i) - u(t_{i-1})| = \sum_{i=1}^n |v_i|(t_i - t_{i-1}) \,.$$

It is observed that $u(t)$ is bounded, i.e., $|u(t)| \le M_u$. Thus, the total variation can be bounded by

$$V_0^T(u) \le M_u \sum_{i=1}^n (t_i - t_{i-1}) = M_u T \,.$$

Consider the spike excitation function, we also can conclude that

$$|s(t_1) - s(t_2)| = |f_e(u(t_1)) - f_e(u(t_2))| \le \frac{u(t_1) - u(t_2)}{u_{\text{firing}}} \,.$$

It is evident that both $u(t)$ and $s(t)$ have finite total variation due to $M_u T < \infty$ and $M_u T / u_{\text{firing}} < \infty$. Therefore, the function expressed by single spiking neuron with the SRM scheme is of bounded variation. Since connection weights is independent to $t$ and bounded by $M_w$, we can further conclude that the function expressed by an SNN with the DEF scheme is the function of bounded variation. This completes the proof. $\qquad \square$