# OpenReview forum: "On the Generalization Bounds of Spiking Neural Networks via Rademacher Complexity"
_ICLR.cc/2026/Conference — ICLR 2026 Conference Withdrawn Submission_

### Official Review · Reviewer_oEUf · 2025-10-27

**Soundness:** 1
**Presentation:** 1
**Contribution:** 1
**Rating:** 2
**Confidence:** 3

**Summary:**

The authors derive generalization bounds for several SNN models by analyzing their Rademacher complexity through covering number arguments.

**Strengths:**

Generalization bounds for SNN models are a highly overlooked topic that deserves more attention. This paper contributes to this area by presenting a variant of an established approach from the learning theory perspective.

**Weaknesses:**

**Clarity**: Many notations are used before they are introduced, and several typographical errors further hinder readability. For example:

- In the main result it is unclear whether $f$ is real-valued and which norm is meant by `$\|f(\cdot)\|$`.
- The phrase “`$N_f$ universally relates to the upper bound of $\|f(\cdot)\|$`” is ambiguous and should be made precise.
- At the point where the main result is stated, neither $\mathcal{D}$ nor the empirical Rademacher complexity has been introduced.
- The relationship between the scalar $x$ and the vector $\mathbf{X}$ in line 60 is not explained.
- In the expression $N_f=\mathcal{O}(TM_w)^L e^{-TL}$ it is not specified with respect to which parameter the asymptotic notation is meant.

These examples illustrate a broader pattern of unclear notation; more instances are listed in the questions section below.

**Mathematical treatment / quality**: From the proofs I reviewed, there appear to be unjustified steps and/or mistakes. For instance, in line 718 the authors write
$$
\int_0^t \|u_{\text{rest}} + \tau_r f_{\text{agg}}(x(\tau))\|  d\tau \leq t  \|u_{\text{rest}} + \tau_r f_{\text{agg}}(x(t))\|,
$$
which is not generally valid without additional assumptions (e.g., constancy of the integrand or a specific relation between $x(\tau)$ and $x(t)$). Such steps should be justified or corrected.

**Lack of proper comparison**: The authors cite the results from *Zhang et al., 2024*, but no direct or meaningful comparison is provided. If a direct comparison is not feasible (e.g., due to differing assumptions or settings), it would still be valuable to contrast the findings with other models or established results. For instance, are the reported bounds or behaviors surprising when compared to what is known for standard ANNs?

In addition, relevant prior work on theoretical bounds is missing. In particular, classical results based on the VC-dimension should be cited, such as:

- Wolfgang Maass and Michael Schmitt. *On the complexity of learning for a spiking neuron.* In *Proceedings of the Tenth Annual Conference on Computational Learning Theory*, pp. 54–61, 1997.
- Michael Schmitt. *VC dimension bounds for networks of spiking neurons.* In *ESANN*, pp. 429–434, 1999.

Another relevant reference that uses covering numbers to bound the generalization gap is:
- A. Martina Neuman, Dominik Dold and Philipp Christian Petersen. *Stable Learning Using Spiking Neural Networks Equipped With Affine Encoders and Decoders.* *arXiv:2404.04549*.

**Questions:**

1. Can the authors clarify whether Eq. (2) applies to all models or only to EID? In either case, it is not clear how one goes from Eq. (2) to line 274. In particular, what does $f$ represent in that context?

2. The sentence in lines 87–88 is unclear. Could the authors please elaborate or rephrase it?

3. The acronym **DEF** appears in line 92, but it is not defined until line 159. Please ensure it is introduced upon first use.

4. In Figure 1, when the authors refer to a “stricter bound for the covering number in Lemma 4,” stricter relative to which bound? Also, note that **DEF** is repeated in the same figure.

5. Please clarify the sentence in lines 232–233. Only DTA appears to have, at least in an obvious manner, the required autoregressive form.

6. In lines 260–263, should it be $TN - f$ or $T N_f$?

7. What do the variables $a$ and $b$ represent in Eq. (3)?

8. How do the authors justify the step made in line 672?

9. In line 683, is $v_i$ a function of $t$? I could not understand the application of the mean value theorem in this context.

10. In the proof of Lemma 3 (Section B.1), the authors seem to use a different definition of functions of bounded variation from the one given in Definition 1, which is otherwise never explicitly used throughout the text.

---

### Official Review · Reviewer_8EZY · 2025-10-29

**Soundness:** 2
**Presentation:** 2
**Contribution:** 1
**Rating:** 2
**Confidence:** 5

**Summary:**

The authors derive a generalisation bound for integrate-and-fire neuron models following the approach presented in (Verma and Kumar, 2025).

While the presented material seems to be technically correct, and I agree with the authors that more theoretical results on generalisation bounds for spiking neural networks (SNNs) are highly relevant, the work has several strong weaknesses that will have to be addressed before it can be published.

In particular, the paper follows one by one the calculation shown in (Verma and Kumar, 2025) for generalisation bounds of neural ODEs, adjusted to the integrate-and-fire neuron. That in itself is not a problem, although it is only mentioned towards the end of the appendix. This should definitely be stated at the beginning of the main paper (e.g., “We adopt the method introduced in … for…”). Moreover, as detailed below, the derived bounds are actually not valid for spiking neurons (unless fully rate-based), contrary to what is claimed in the paper. This has to be framed correctly and thoroughly discussed in the paper.

**Strengths:**

Using some of the techniques from (Verma and Kumar, 2025) for SNNs is interesting.

**Weaknesses:**

1. The authors claim to be the first to derive generalisation bounds for SNNs using covering numbers. This has been done before for simplified spiking neuron models in (Neuman, Dold & Petersen, arXiv:2404.04549,  2024) for cases where output spike times depend continuously on inputs and parameters.
2. The authors do not consider standard spike mechanisms. In line 166, they introduce their activation mechanism, which simply rescales the membrane potential by the value of the firing threshold. Hence, neurons communicate using continuous values, not spikes. As far as I can see, the reset mechanism is also not considered in the derivation.
3. In their derivation, to obtain covering numbers, it is, several times, a requirement that the output of their SNN is Lipschitz continuous with respect to parameters and inputs. This is, however, not the case for actual spikes (for instance, line 736 and line 696 only work for non-spiking activations), which can jump, disappear, or reappear depending on their inputs (see Klos & Memmesheimer, Phy. Rev. Letter 134, 2025). Thus, I doubt the approach can be generalised to actual LIF dynamics.
4. Similarly, the calculations are heavily adopted from (Verma and Kumar, 2025), and the derived bounds are very similar to the results reported in (Verma and Kumar, 2025).
5. The derived bounds are shown to scale exponentially with the number of layers L and integration time steps T, leading to vanishing bounds for large enough values of L or T. This only occurs because N_f, the value range of SNN outputs, is set to be bounded by the absolute value of a similarly scaling term (and N_f occurs in the bound). Thus, in the limit of large L or T, the SNN only outputs 0, leading to perfect generalisation since the SNN does not learn anything. This is a trivial result.
6. The paper has quite a few typos, repeated words, etc.

**Questions:**

1. If you have the covering number, you can directly calculate generalisation bounds without Rademacher complexity. Why are you doing this extra step?
2. In the appendix, line 738, I believe the upper index is missing for u(0). Similarly, should l be L instead?
3. In the appendix, line 683: the notation for d/dt v seems incorrect?

---

### Official Review · Reviewer_Mnr1 · 2025-10-31

**Soundness:** 3
**Presentation:** 3
**Contribution:** 2
**Rating:** 6
**Confidence:** 3

**Summary:**

This work proposes the generalization bounds for typical LIF spiking neurons via empirical Rademacher complexity and covering number. The generalization bound with exponential reduction with respect to maximum duration T and network depth L is tighter than previous work. An experiment on delayed-memory XOR task is conducted to evaluate the effectiveness of the theoretical results.

**Strengths:**

1. Novelty. The author attempts to investigate the generalization ability of the most common integration-and-fire mechanism in SNNs which represents a pioneering and original contribution.

2. The structure of the proof is well-organized and rigorous. The author briefly illustrates the components of the proof with a single diagram.

**Weaknesses:**

1. By directly addressing the problem the paper aims to solve without providing sufficient background on generalization theory, the readability of the paper is reduced. Since I am not an expert in the field of machine learning generalization, I had to make considerable effort to understand the theorem presented in the introduction and its significance. (See questions)

2. The paper does not mention the general approaches, related work, or progress in analyzing the generalization error of neural networks in the ANN field. This makes it difficult for me to assess the importance, contribution, and impact of this work for SNNs.

3. The experiments are not sufficiently clear. For example, the authors do not provide details about the architecture of the neural networks used or their training procedures. The experimental conclusions only offer a qualitative analysis of the relationships between generalization error, network depth, width, and maximum duration, but lack quantitative connections to the upper bound presented in the theorem. In addition, the legend and font size in Figure 3 are not clear enough.

**Questions:**

1. The author provides the definition of Rademacher complexity in the main text. However, what is the role and significance of Rademacher complexity in generalization theory? Why is this particular complexity measure used instead of other notions of complexity? Due to my limited background knowledge in this area, I found the introduction of Rademacher complexity rather abrupt.

2. what is \alpha in the Rademacher complexity upper bound of Theorem 1?

3. Could the author briefly introduce the existing work on theoretical characterizations in the SNNs field? This would help clarify the role of generalization theory within these theoretical analyses.

---

### Official Review · Reviewer_7yAX · 2025-11-01

**Soundness:** 3
**Presentation:** 2
**Contribution:** 3
**Rating:** 6
**Confidence:** 3

**Summary:**

Authors derive analytical bound for spiking neural networks using Rademacher complexity and covering numbers. The bound is informative on how the model performance generalises from training to testing data. It ensures that the error on testing data is bounded via the training error and model complexity, evaluated as Rademacher complexity. The paper builds on previous results that bounds the test error of networks with stochastic firing, but differs significantly from this previous work in its scope of addressing spiking networks in general. The bulk of analytical work is accompanied by small numerical experiments.

**Strengths:**

The question addressed is interesting and the results are potentially insightful. While the paper builds on previous results, it brings important new results that are original and seem significant. The derivation appears technically correct, even though I was not able to check all the details. The relation to closest previous work (Zhang et al. 2024) is sufficiently outlined.

**Weaknesses:**

1) The paper is difficult to read and could profit a lot from introducing intuitive introduction to the methods and intuitive explanations of results. For example, before stating the Theorem 1, it would be very useful to give some intuition about it, because it is not even clearly stated what does the theorem want to achieve. This becomes clear from further reading, but it would be better to clearly explain it before stating the theorem. Also, the bound on the Rademacher complexity could be better explained. For example, it could be explained why and how the bound depends on certain parameters. An intuitive explanation of conditions that have to be satisfied for the Theorem 1 to hold would be very useful, too.

2) Limitations are unfortunately not discussed. Could authors add a section where they discuss limitations of their study?

3) The text contains typos and grammatically awkward sentences. There is a certain number of them, even thought the paper seems in general written with some care. Below, I listed a few examples, but there are more. I advise to carefully revise the text for such textual mistakes.
Examples:
-line [20] last sentence of the abstract does not make much sense to me, present paper cannot shed insights into future studies.
-line [25] "have been attracted" is grammatically incorrect
-line [46-47] " It is potential to reduce the generalization bound of SNNs ..." does not seem a correct sentence.
-line [450] please use showed instead of "showd",
-line [044] The sentence containing " the generalization of SNNs with general spiking neurons" is unclear.

4) The Figure 3  is visually unappealing and, more importantly, the font on the axis is so small that is almost unreadable. Further, the x-axis on Figure 3, right, reads "The number of network layer $N_w$ while I think it should be "network width $N_w$". To improve clarity, figure legends should specify the variable and not just its value, e.g. T=500, T=1000, T=1500, etc. Finally, the figure caption is unclear and should be rewritten.

**Questions:**

1) [line 56] Why is Lipschitz condition necessary? It seems to be a strong condition.

2) [lines 422-423] mention training and testing errors. How are these errors defined and estimated in the numerical experiments?

3) [421-422] Why are quantities T=[500 : 500:3000], N_w=[2:2:8], expressed in brackets?

4) I presume that the generalisation capacity of a network might depend on the sparsity of the connectivity and on the width of the distribution of the connectivity weights. Can authors speculate about it?

6) A recent study in [1] computed analytical bounds on firing rates and the representational error for efficient spiking networks. Could authors comment on the relation to their current results?

The loss in efficient spiking networks is formulated in such a way as to bound the representational error (see e.g. [1]). If the number of variables that the network represents is smaller than the number of neurons, the solutions are degenerate, in the sense that there can be many spiking patterns that can similarly well approximate the target. In other words, very different spiking patterns can give a similar population readout $\hat{x}$ and thus a similar (training) error $\epsilon = (x(t)-\hat{x}(t))^2$. Can authors speculate what might be consequence of such degeneracy of solutions on Rademacher complexity and generalization bounds?

[1] Urdu, Matin, et al. "Firing Rates and Representational Error in Efficient Spiking Networks Are Bounded by Design." International Conference on Artificial Neural Networks. Cham: Springer Nature Switzerland, 2025.

---

### Note · Authors · 2025-11-25

I have read and agree with the venue's withdrawal policy on behalf of myself and my co-authors.